# Evaluation of γ-oryzanol Accumulation and Lipid Metabolism in the Body of Mice Following Long-Term Administration of γ-oryzanol

**DOI:** 10.3390/nu11010104

**Published:** 2019-01-06

**Authors:** Eri Kobayashi, Junya Ito, Naoki Shimizu, Takumi Kokumai, Shunji Kato, Kazue Sawada, Hiroyuki Hashimoto, Takahiro Eitsuka, Teruo Miyazawa, Kiyotaka Nakagawa

**Affiliations:** 1Food and Biodynamic Chemistry Laboratory, Graduate School of Agricultural Science, Tohoku University, Sendai, Miyagi 980-8572, Japan; eri.k@dc.tohoku.ac.jp (E.K.); junyai@tohoku.ac.jp (J.I.); nshimizu@dc.tohoku.ac.jp (N.S.); t.koku@dc.tohoku.ac.jp (T.K.); takahiro.eitsuka.a1@tohoku.ac.jp (T.E.); 2Department of Cell Biology, Division of Host Defense Mechanism, Tokai University School of Medicine, Isehara, Kanagawa 259-1193, Japan; tyunkato@gmail.com; 3Tsuno Food Industrial CO., LTD., Ito-Gun, Wakayama 649-7194, Japan; sawada.kazue@tsuno.co.jp (K.S.); hirohas@tsuno.co.jp (H.H.); 4Food and Biotechnology Innovation Project, New Industry Creation Hatchery Center (NICHe), Tohoku University, Sendai, Miyagi 980-8579, Japan; miyazawa@m.tohoku.ac.jp; 5Food and Health Science Research Unit, Graduate School of Agricultural Science, Tohoku University, Sendai, Miyagi 980-8572, Japan

**Keywords:** γ-oryzanol, absorption and metabolism, HPLC-MS/MS, rice bran oil

## Abstract

γ-Oryzanol (OZ), abundant in rice bran oil, has gained attention due to its physiological activities (e.g., lipid-lowering effects). However, the absorption and metabolism of orally ingested OZ have not yet been fully elucidated. In this study, diets containing normal or high contents of OZ were fed to obesity model mice for 8 weeks, and OZ concentrations in plasma and organs were analyzed by HPLC-MS/MS. To evaluate the relationship between OZ accumulation and lipid metabolism in vivo, lipid concentrations in the mice plasma and liver were also measured. As a result, the accumulation of intact OZ in plasma and organs was seen in mice fed diets containing OZ, where mice fed diets containing higher OZ contents demonstrated higher levels of OZ accumulation and lower amounts of plasma lipids. These results, in combination with our additional data from a single oral administration test, suggest the possibility that intact OZ, along with its metabolites (e.g., ferulic acid), is biologically-active.

## 1. Introduction

Functional substances contained in plants such as cereals, vegetables and fruits are presumed to possess various bioregulatory functions. To fully receive such functions, it is necessary to understand how these functional substances are absorbed and metabolized in vivo. However, studies regarding the absorption and metabolism of such substances remain limited, with the exception of a few reports on some well-known molecules (e.g., tocopherols, polyphenols, and certain carotenoids [1,2,3].

γ-Oryzanol (OZ), which has gained popularity in recent times, is also one such functional substance. OZ, a mixture of ferulic acid esters of plant sterols and triterpene alcohols, is abundant in rice bran and rice bran oil. Cycloartenyl ferulate, 24-methylenecycloartanyl ferulate, campesteryl ferulate, and β-sitosteryl ferulate are the most common components of OZ in rice bran, and in addition to these, the putative isomers of 24-methylenecycloartanyl ferulate are contained in rice bran oil (Figure 1) [4]. OZ has been associated with various beneficial effects, including lipid-lowering effects [5,6,7,8,9,10]. Furthermore, in Japan, OZ has been used as a drug for health purposes [11]. Hence, there is considerable interest in the beneficial effects of OZ on human health; however, the absorption and metabolism of orally-administered OZ have not yet been fully characterized.

In 1980, Fujiwara et al. investigated the metabolic pathway of OZ by oral administration of 14C-labeled OZ (200 mg/kg body weight) to rabbits [12]. As a result, 80% of the radioactivity in blood was derived from a single metabolite, namely ferulic acid, rather than OZ itself, suggesting that OZ is hydrolyzed after absorption, and exists mainly as ferulic acid in vivo. Since this finding, subsequent studies have essentially focused on ferulic acid, rather than OZ. Contrary to these previous findings, by utilizing HPLC-MS/MS, we recently found that the OZ species exist in their intact form in mice plasma after a single oral administration of OZ [4]. From this result, we hypothesized that part of the ingested OZ is directly absorbed into the blood stream, and exerts physiological actions (e.g., lipid-lowering effects) in vivo in the intact form.

To further verify this hypothesis, in this study, we fed two types of rice bran oil containing different concentrations of OZ (Figure 2) to obesity model mice (i.e., fetal programming model [13,14,15]), and analyzed the OZ concentrations of plasma and organs by HPLC-MS/MS. We further measured lipid parameters in the plasma and liver to evaluate the relationship between the presence of OZ in the plasma and liver and the physiological effects of OZ.

## 2. Materials and Methods

### 2.1. Materials and Diets

Soybean oil (SO) was purchased from Wako Pure Chemical Industries, Ltd. (Osaka, Japan). Nomal rice bran oil (RBO) and rice bran oil containing a high concentration of OZ (HOZ) were provided from Tsuno Food Industrial Co., Ltd. (Wakayama, Japan). RBO and HOZ contained about 0.17% and 1.36% OZ respectively, quantified based on a modified ultraviolet (UV) spectrophotometric (325 nm) method [4]. The AIN93G premix, containing all AIN93G ingredients except for corn starch and fat source, was obtained from Research Diets, Inc. (New Brunswick, New Jersey, USA.). Test diets were AIN93G (containing 7% (*w*/*w*) fat as SO) and three types of high-fat diets; SO-HF (containing 20% (*w*/*w*) SO), RBO-HF (containing 20% (*w*/*w*) RBO), and HOZ-HF (containing 20% (*w*/*w*) HOZ). High-fat diets were prepared from the AIN93G premix, to which 20% test oil and corn starch (W-LIP, JAPAN CORN STARCH CO., Ltd., Tokyo, Japan) for weight adjustment were added. Details of the diet composition are shown in Table 1.

### 2.2. Animal Experiment of Obesity Model (Fetal Programming Model) Mice 

C57BL/6J dams in their third day of pregnancy were purchased from CLEA Japan (Tokyo, Japan). Dams were housed individually in polycarbonate cages with free access to food and distilled water in a room at constant temperature (23 ± 1 °C) and humidity under a 12 h light/dark cycle. During pregnancy and lactation, the dams were fed with the SO-HF diet. Each dam gave birth to 5–9 pups. After birth, seven pups were randomly assigned to each dam, and weaned at 3 weeks of age. At 3 weeks of age, male offspring were randomly separated into four dietary groups (*n* = 9) and freely fed test diets (AIN93G, SO-HF, RBO-HF, and HOZ-HF diets) for 8 weeks. Mice were fasted overnight (14 h) and then sacrificed by decapitation at 11 weeks of age, and their liver, brain, kidney, spleen, muscle, mesenteric fat, perirenal fat, epidydimal fat, and blood were collected. Plasma was prepared from blood via centrifugation at 1000× *g* for 20 min at 4 °C. All decapitation was performed under isoflurane anesthesia, and all efforts were made to minimize suffering. Animal experiments were carried out in strict accordance with the Regulations for Animal Experiments and Related Activities at Tohoku University. The protocol was approved by the Center for Laboratory Animal Research, Tohoku University (Permit Number: 2015AgA-028).

### 2.3. Extraction of OZ from Plasma and Organs

Total lipids were extracted from plasma and organs by the Folch method [4,13,16]. Plasma (100 μL) was diluted with 500 μL of 0.9% KCl aqueous solution, and each organ (200 mg) was homogenized with four volumes of 0.16 M saline (1 mM ethylenediamine tetraacetic acid). To the diluted plasma (600 µL) and organ homogenate (400 μL) were added 2.4 mL and 1.6 mL chloroform–methanol (2:1, *v*/*v*) containing 0.002% butylated hydroxytoluene, respectively. The extract was partitioned by centrifugation at 2000× *g* for 20 min at 4 °C into two layers: the chloroform layer (lower organic layer) and the methanol–water layer (upper layer). The lower chloroform layer (lipid fraction) was collected. The remaining aqueous layer containing a semi-solid interface was re-extracted using Folch’s theoretical lower phase and was subjected to centrifugation at 2000× *g* for 20 min at 4 °C. The lower organic layer was collected and combined with the previously extracted lower layer. The combined lipid fraction was rinsed with Folch’s theoretical upper phase and was evaporated under nitrogen gas. The dried plasma extract was redissolved in 600 µL methanol and a 10 μL aliquot was subjected to HPLC-MS/MS analysis for OZ analysis. The dried organ extracts were redissolved in 1 mL hexane–chloroform (9:1, *v*/*v*), and 500 µL of this mixture was loaded onto a Strata^®^ SI-1 Silica cartridge (Phenomenex Inc., Torrance, CA, U.S.A.) equilibrated with hexane–chloroform (9:1, *v*/*v*). The cartridge was rinsed with 1.5 mL hexane–chloroform (9:1, *v*/*v*) and OZ was eluted with 1.5 mL hexane–2-propanol (7:3, *v*/*v*). The eluent was evaporated, and the residue was dissolved in 200 µL methanol. A 5 or 10 µL final aliquot was subjected to HPLC-MS/MS analysis.

### 2.4. HPLC-MS/MS analysis of OZ

Standard cycloartenyl ferulate was purchased from Wako Pure Chemical Industries, Ltd. (Osaka, Japan). Other OZ standards (i.e., 24-methylenecycloartanyl ferulate, campesteryl ferulate, and β-sitosteryl ferulate) were fractionated and purified from rice bran. Before analyzing the plasma or organ samples, standard OZ solutions (0.06–600 μg/mL) were prepared by diluting each standard in methanol to its expected level contained in samples. Then, 10 μL of each standard solution was subjected to HPLC-MS/MS analysis to construct a standard curve. The HPLC-MS/MS system consisted of a Shimadzu liquid chromatography system (Shimadzu, Kyoto, Japan) and a 4000 QTRAP mass spectrometer (SCIEX, Tokyo, Japan). OZ standards and biological samples were analyzed using a C18 column (COSMOSIL 2.5C18-MS-II, 2.5 μm, 2.0 ID × 100 mm; Nacalai Tesque, Inc., Kyoto, Japan) with a binary gradient consisting of solvent A (methanol–acetic acid (99:1, *v*/*v*)) and solvent B (2-propanol). The gradient profile was as follows: 0–4.5 min, 0% B; 4.5–4.6 min, 0–100% B linear; 4.6–6.6 min, 100% B; 6.6–6.7 min, 100–0% B linear; 6.7–8.7 min, 0% B. The flow rate was 0.5 mL/min, and the column temperature was 40 °C. OZ was detected by the multiple reaction monitoring (MRM) transitions either defined in the literature or predicted MRM ion pairs (Appendix A).

### 2.5. Biochemical Parameters

Plasma triglyceride (TG), total cholesterol (TC), phospholipid (PL), and glucose levels, as well as liver TG and TC levels, were measured using commercial enzyme kits (Wako Pure Chemical Industries, Ltd., Osaka, Japan) according to the manufacturer’s protocol. Liver PL levels were determined using the method of Rouser [13,17].

### 2.6. Single Oral Administration of OZ to Rats

Male Sprague–Dawley rats were purchased from CLEA Japan. Rats were housed individually under conditions described above. After 18 h fast, rats (*n* = 4) received OZ (containing cycloartenyl ferulate, 24-methylenecycloartanyl ferulate, campesteryl ferulate, and β-sitosteryl ferulate) (181 mg/kg) dissolved in soybean oil (55.9 g/kg) via gastric intubation. After 6 h of administration, blood was collected, and plasma was prepared from blood via centrifugation. Plasma (80 μL) was subjected to extraction and analysis of ferulic acid. The remaining plasma (20 μL from each rat) was combined and subjected to extraction and analysis of OZ described above. The protocol of this study was approved by the Center for Laboratory Animal Research, Tohoku University (Permit Number: 2018AgA-040).

For extraction of ferulic acid and plasma (80 μL) was diluted in 400 μL of methanol containing 1% formic acid, vortexed, and subjected to centrifugation at 13,000× *g* for 10 min at 4 °C. The upper layer was collected, and the residue was similarly re-extracted with 300 μL of methanol containing 1% formic acid. The upper layers were combined and evaporated under nitrogen gas. The dried extract was redissolved in 1.5 mL of methanol–water (1:9, *v*/*v*) containing 0.1% formic acid. This mixture was loaded onto an Oasis HLB cartridge (Waters, Milford, MA, USA.) equilibrated with 0.1% formic acid aqueous solution. The cartridge was rinsed with 1.5 mL of 0.1% formic acid aqueous solution and ferulic acid was eluted with 1.5 mL methanol–water (9:1, *v*/*v*) containing 0.1% formic acid. The eluent was evaporated, dissolved in 1 mL acetonitrile–water (1:9, *v*/*v*), and subjected to HPLC-MS/MS analysis described below.

Samples were analyzed with a 4000 QTRAP mass spectrometer (SCIEX) with a C8 column (InertSustain C8, 2 μm, 2.1 ID × 100 mm; GL Science, Tokyo, Japan) maintained at 40 °C with a flow rate of 0.4 mL/min. Gradient elution was performed using a two-solvent system consisting of solvent A (water containing 0.1% formic acid) and solvent B (acetonitrile containing 0.1% formic acid). The gradient program was as follows: 0–3.5 min, 10–20% B linear; 3.5–5.5 min, 20–40% B linear; 5.5–6.0 min, 40% B. Ferulic acid was detected with MRM ion pairs described in Appendix A.

### 2.7. Statistics

Data are expressed as the means ± standard errors (SE). Data regarding the OZ concentration in plasma and organs were analyzed by the two-tailed unpaired Welch’s t-test. Differences were considered significant at *p* < 0.05 or *p* < 0.01. Data regarding the plasma and liver parameters were analyzed by one-way ANOVA followed by Tukey’s test. Differences were considered significant at *p* < 0.05.

## 3. Results and Discussion

### 3.1. Body and Organ Weight of Fetal Programming Model Mice

In a previous study, we found that OZ exists in the intact form in mice plasma after a single oral administration of OZ. We thus considered the possibility that OZ in the intact form, rather than in the form of ferulic acid as previous studies point out, is responsible for the well-known physiological properties of OZ. In this study, to verify the above hypothesis, we examined the relationship between the physiological functions of OZ (e.g., improvement of lipid metabolism and fat accumulation suppression) and the accumulation of OZ in vivo. As an obesity model, fetal programming model mice, prepared based on the Developmental Origins of Health and Disease (DOHaD) hypothesis which suggests that maternal malnutrition during pregnancy and lactation increase the risk of metabolic syndrome in offspring [18], were adopted. For instance, previous studies have suggested that maternal high fat diet induces obesity in pups [13,14], and hence, we considered that this model would be useful for investigating the influence of OZ accumulation on lipid metabolism.

In this study, diet composition did not affect the body weight and food intake during the breeding period (Table 2). Organ weights other than white adipose tissue were not affected by diet composition (Table 2). Compared to the AIN93G group, the perirenal adipose tissue weight tended to increase in the SO-HF group (Table 2, *p* = 0.15), which confirmed our consideration that the model was suitable for investigating the relationship between OZ accumulation and lipid metabolism. Compared to the SO-HF group, the RBO-HF group and especially the HOZ-HF group demonstrated low perirenal adipose tissue weight (Table 2). A similar tendency was also observed in the epidydimal fat tissue weight (Table 2). These results demonstrate that the intake of rice bran oil (i.e., RBO and HOZ, especially HOZ) prevents the growth of white adipose tissue and enhances lipid metabolism in mice. The fact that the effect of the HOZ diet was stronger than that of the RBO diet also suggests the involvement of OZ in the above phenomena. These results agree with those of a previous study where OZ treatment to mice decreased the size of white adipose tissue and prevented the formation of fat [19]. Although accumulation of OZ was observed in mesenteric white adipose tissue, which is also a visceral adipose tissue, no significant difference in mesenteric white adipose tissue weight was seen. Such difference is of interest, and further studies may be necessary to elucidate its mechanism.

### 3.2. OZ Concentration in Plasma and Organs

In a previous study, we developed a method to analyze the molecular species of OZ with the use of HPLC-MS/MS and evaluated its use in the analysis of biological samples [4]. Based on this method, in this study, the analytical conditions were optimized to achieve a more rapid and accurate measurement of OZ. In the MRM chromatograms, clear peaks of cycloartenyl ferulate, 24-methylenecycloartanyl ferulate, campesteryl ferulate, and β-sitosteryl ferulate standards were detected at 3.3, 3.5, 3.7, and 4.0 min, respectively (Figure 3a). Detection limits were 0.6 pg per injection at a signal-to-noise ratio of 3.

In our previous study, OZ species were detected in their intact form in mice plasma after a single oral administration of OZ [4]. Similarly, intact OZ were detected in the plasma of OZ administered mice (RBO-HF and HOZ-HF groups) in this study (Figure 3B and Figure 4A). The OZ concentration in plasma was significantly higher in the HOZ-HF group (cycloartenyl ferulate 18.6 ng/mL, 24-methylenecycloartanyl ferulate isomers (i.e., 24-methylenecycloartanyl ferulate, cyclobranyl ferulate, and cyclosadyl ferulate) 20.5 ng/mL, campesteryl ferulate 10.7 ng/mL, and β-sitosteryl ferulate 2.9 ng/mL) than the RBO-HF group (Figure 4A), and thus confirmed that a significant amount of OZ remains in the plasma after a long-term feeding of OZ. These concentrations were similar to those of our previous study in which a single oral administration of OZ was performed, despite the fact that the OZ intake in the HOZ-HF group (295 mg/kg B.W./day) was about half the amount of OZ that was administrated in our previous study (600 mg/kg B.W.). The possible reason why the same concentration of OZ was detected in spite of the difference in the intake of OZ may be that OZ remains in the blood for a long period, or is re-released from the organs. We therefore analyzed the OZ species in organs by HPLC-MS/MS.

OZ was detected from all organs (liver, brain, kidney, spleen, muscle, mesenteric fat, and perirenal fat) in the RBO-HF and HOZ-HF groups, where the OZ concentration was significantly higher in the HOZ-HF group than the RBO-HF group (Figure 3C and Figure 4B–H). Especially, the OZ content was remarkably high in the liver (cycloartenyl ferulate 2.3 μg/g, 24-methylenecycloartanyl ferulate isomers 1.6 μg/g, campesteryl ferulate 0.3 μg/g, and β-sitosteryl ferulate 0.1 μg/g, Figure 4B), whereas only small amounts were detected from the brain (cycloartenyl ferulate 2.4 ng/g, 24-methylenecycloartanyl ferulate isomers 2.9 ng/g, campesteryl ferulate 1.4 ng/g, and β-sitosteryl ferulate 0.6 ng/g, Figure 4C). OZ was also found to accumulate in the other organs, which suggest that OZ, once ingested, is widely distributed in the mice body (Figure 4D–H). Previous studies have reported that after administration of ^14^C-OZ, radioactivity was observed mainly in the liver [20]; the fact that significant amounts of OZ were accumulated in the liver in the present study suggests the possibility that this radioactivity was in fact derived from intact OZ. In addition, while cycloartenyl ferulate was found to accumulate in the liver at a high concentration, 24-methylenecycloartanyl ferulate isomers were found at relatively high concentrations in the plasma, which suggests that there may be a difference in the metabolism or accumulation between the OZ molecular species. With regard to the presence of OZ in the brain, the movement of OZ to the brain has been investigated in past studies, but there has been consensus on the cause. In such studies, radioactivity was hardly observed from the brain after a single oral administration of ^14^C-OZ, but on the other hand, a remarkably higher radioactivity was observed after long-term administration of ^14^C-OZ [21]. Furthermore, in a different study using UV-HPLC, a relatively high UV absorption at 315 nm, corresponding to OZ, was detected from the brain 1 h after a single administration of OZ [22]. In this study, the OZ concentration in the brain was low compared to other organs, which suggests that OZ in the intact form accumulates in very small amounts in the brain. In this study, with the use of HPLC-MS/MS, we demonstrated, for the first time, the distribution of intact OZ in organs after long-term feeding on an OZ-rich diet. Intact OZ existed in the plasma and the liver, especially at high concentrations; thus, we measured lipid parameters in the plasma and liver to evaluate the physiological effects of OZ.

### 3.3. Plasma and Liver Parameters of the Mice

To evaluate the relationship between the presence of OZ and its physiological effects, we measured the biological parameters of the plasma and the liver, in which the concentrations of OZ were higher than in other organs. The plasma TG in the HOZ-HF group was significantly lower compared with the RBO-HF group (Table 2). The plasma TC and PL showed a similar tendency (Table 2). There was no significant difference in the plasma glucose and liver lipids (TG, TC, and PL) among the four groups (Table 2). These results, specific to the HOZ-HF group, suggest that OZ reduces plasma lipid levels in mice, since there is little difference between the RBO-HF and HOZ-HF diets except for the OZ content. Many human and rodent studies have demonstrated that OZ administration causes blood lipid reduction [9,23,24]. In these studies, it was suggested that the ingested OZ was first metabolized, and the resultant metabolites (e.g., ferulic acid and plant sterols) are responsible for the bioactivity. In contrast, the result of this study, in which the HOZ-HF group demonstrated a significantly high concentration of plasma OZ and a decrease in plasma lipid concentrations, implies a correlation between the concentration of OZ and lipid-lowering effects in plasma. This may suggest that intact OZ is biologically active in vivo. In fact, in vitro tests have also suggested that OZ, in the intact form, demonstrates hypocholesterolemic activities [25]. However, this consideration acknowledges the activity of OZ metabolites (e.g., ferulic acid). As a result of analyzing OZ and ferulic acid in plasma after a single administration of OZ to rats, it was found that the plasma concentrations of OZ and ferulic acid were similar; the total OZ level in plasma was 43.4 nM (cycloartenyl ferulate 16.4 ng/mL (27.1 nM), 24-methylenecycloartanyl ferulate 7.3 ng/mL (11.7 nM), campesteryl ferulate 1.9 ng/mL (3.3 nM), and β-sitosteryl ferulate 0.8 ng/mL (1.3 nM)), whereas that of ferulic acid was 36.1 ± 12.0 nM. Consequently, these results suggest the possibility that intact OZ, as well as its metabolites (e.g., ferulic acid), are biologically-active forms that exhibit physiological actions. Further studies, including the analysis of metabolites such as ferulic acid and sterols, are necessary to prove our hypothesis. Since intact OZ was also accumulated in plasma, liver, and other organs (Figure 4), further examination of these organs (e.g., fat and kidney) may be also required to understand the relationship between the accumulation of intact OZ, along with its metabolites, and their physiological effects in vivo.

## 4. Conclusions

In conclusion, intact OZ was detected in mice plasma and organs after a long-term administration of OZ (RBO-HF and HOZ-HF groups), suggesting that OZ, after being ingested, is widely distributed in the body. Furthermore, higher intake of OZ was associated with higher OZ levels in the plasma and organs, and mice that were fed the HOZ-HF diet demonstrated lower plasma lipid concentrations than the those fed a RBO-HF diet. These results suggest that intact OZ, along with its metabolites, is biologically active, exhibiting physiological actions (e.g., blood lipid-lowering effects). However, to verify this possibility, further studies including the analysis of metabolites such as ferulic acid and plant sterols are necessary, and thus, we are currently performing the analysis of these compounds. Our current study, along with other such studies, are expected to contribute to the elucidation of the absorption and metabolism of OZ in vivo, as well as of the mechanisms by which the physiological actions of OZ are exhibited.

## Figures and Tables

**Figure 1 nutrients-11-00104-f001:**
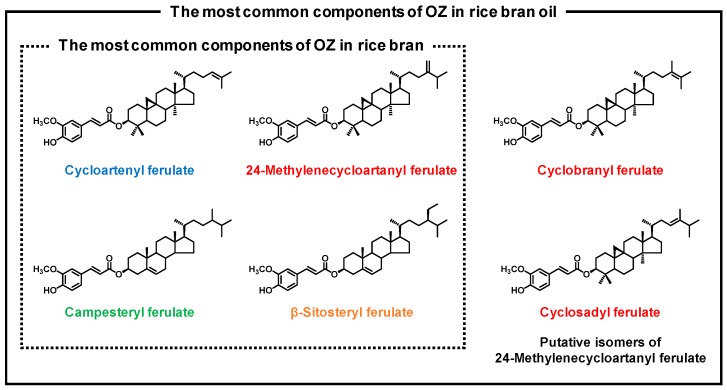
Chemical structures of the most common components of OZ in rice bran and rice bran oil: cycloartenyl ferulate, 24-methylenecycloartanyl ferulate, campesteryl ferulate, and β-sitosteryl ferulate. Cyclobranyl ferulate and cyclosadyl ferulate are possible isomers of 24-methylenecycloartanyl ferulate.

**Figure 2 nutrients-11-00104-f002:**
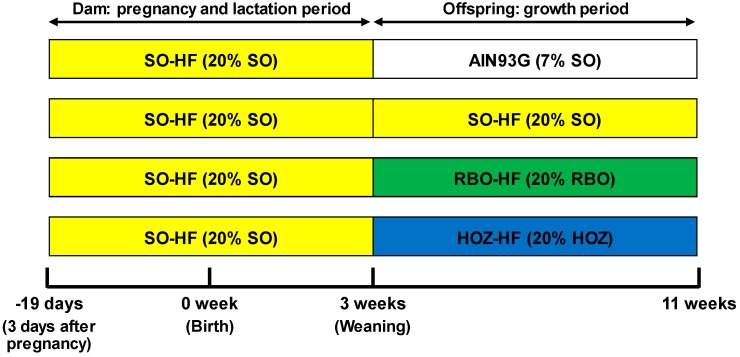
Animal study design. C57BL/6J mouse dams were fed SO-HF diets during pregnancy and lactation periods. All offspring were weaned at 3 weeks of age. Male offspring were fed one of the four diets (AIN93G, SO-HF, RBO-HF, or HOZ-HF) for 8 weeks and were sacrificed at 11 weeks of age. SO: soy bean oil, RBO: rice bran oil, HOZ: rice bran oil containing a high concentration of OZ, AIN93G: a diet containing 7% (*w*/*w*) SO as a fat source, SO-HF: a high fat diet containing 20% (*w*/*w*) SO as a fat source, RBO-HF: a high fat diet containing 20% (*w*/*w*) RBO as a fat source, HOZ-HF: a high fat diet containing 20% (*w*/*w*) HOZ as a fat source.

**Figure 3 nutrients-11-00104-f003:**
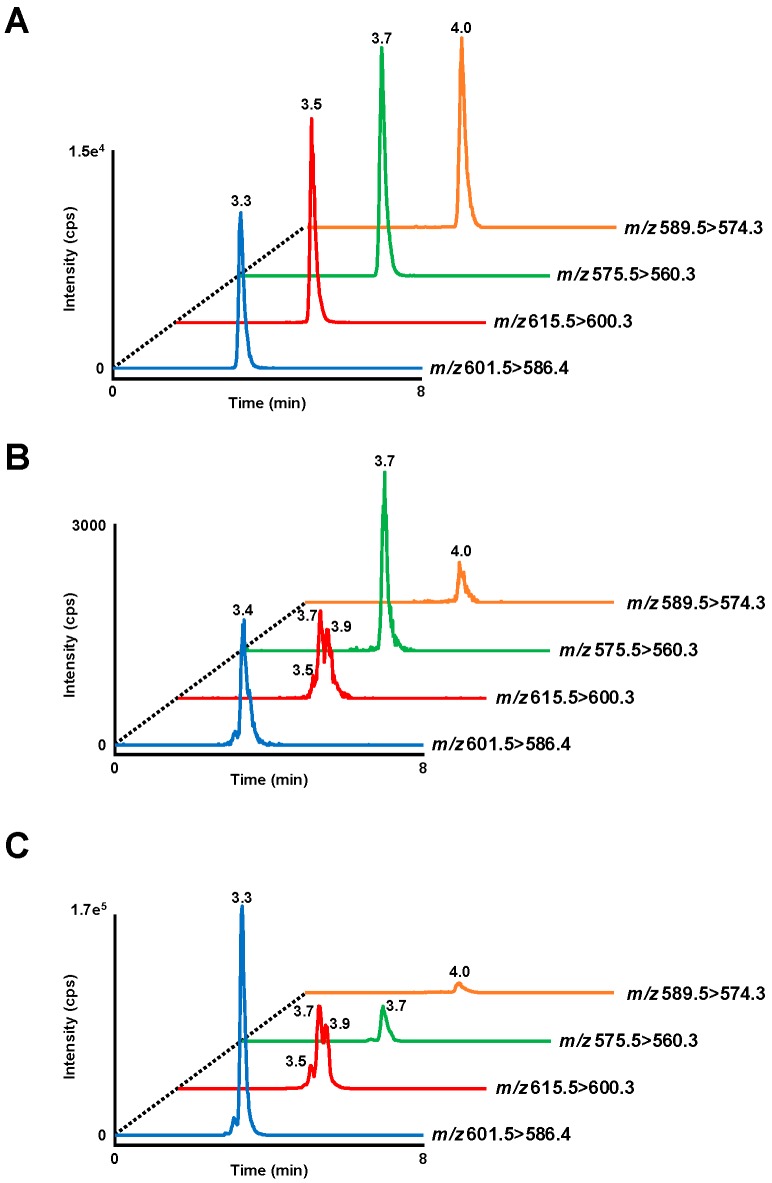
MRM chromatograms obtained by injecting a mixture containing 60 pg each cycloartenyl ferulate (m/z 601.425 [M − H]^−^), 24-methylenecycloartanyl ferulate (m/z 615.441 [M − H]^−^), campesteryl ferulate (m/z 575.410 [M − H]^−^), and β-sitosteryl ferulate (m/z 589.425 [M − H]^−^) standards (**A**). Representative MRM chromatograms of cycloartenyl ferulate, 24-methylenecycloartanyl ferulate isomers, campesteryl ferulate, and β-sitosteryl ferulate in mouse plasma (**B**) or liver (**C**) of the HOZ-HF group. A 5 μL plasma or liver extract was injected into the HPLC-MS/MS. Detailed analytical conditions are described in the Materials and methods section.

**Figure 4 nutrients-11-00104-f004:**
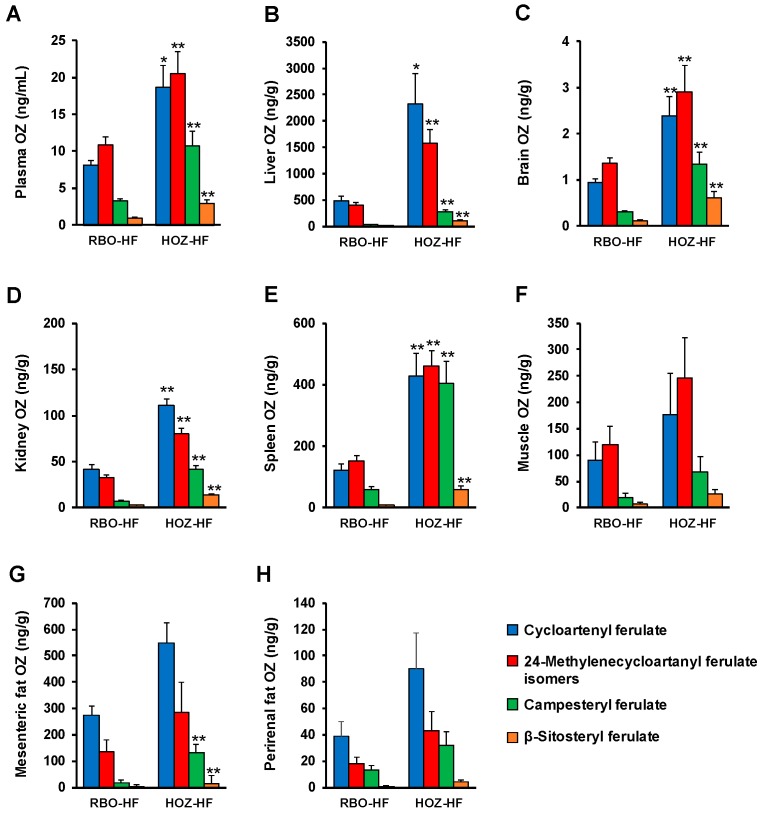
OZ concentration of plasma (**A**) and organs; liver (**B**), brain (**C**), kidney (**D**), spleen (**E**), muscle (**F**), mesenteric fat (**G**), and perirenal fat (**H**). Means ± SE (*n* = 8–9), * *p* < 0.05, ** *p* < 0.01, vs. RBO-HF.

**Table 1 nutrients-11-00104-t001:** Diet composition.

	AIN93G	SO-HF	RBO-HF	HOZ-HF
	**Ingredients (g %)**			
Casein	20.0	20.0	20.0	20.0
L-Cystine	0.3	0.3	0.3	0.3
Corn Starch	39.7	26.7	26.7	26.7
Maltodextrin 10	13.2	13.2	13.2	13.2
Sucrose	10.0	10.0	10.0	10.0
SO	7.0	20.0	-	-
RBO	-	-	20.0	-
HOZ	-	-	-	20.0
Cellulose, BW200	5.0	5.0	5.0	5.0
Mineral Mix S10022G	3.5	3.5	3.5	3.5
Vitamin Mix V10037	1.0	1.0	1.0	1.0
Choline bitartrate	0.3	0.3	0.3	0.3
*t*-Butylhydroquinone	0.0014	0.0014	0.0014	0.0014
Total	100.0	100.0	100.0	100.0
	**Centesimal Composition (kcal %)**			
Protein	20	17	17	17
Carbohydrate	64	44	44	44
Fat	16	39	39	39
kcal/g	4.4	5.1	5.1	5.1

**Table 2 nutrients-11-00104-t002:** Body weight, food intake, organ weights, and biological parameters of plasma and liver.

	Groups			
	AIN93G	SO-HF	RBO-HF	HOZ-HF
Body weight (g)	28.0 ± 0.5	29.5 ± 1.0	28.5 ± 0.5	27.7 ± 0.7
Food intake (kcal/day)	13.5 ± 0.2	13.0 ± 0.1	12.6 ± 0.5	13.8 ± 0.1
Liver (g/100 g B.W.)	3.80 ± 0.14	3.28 ± 0.15	3.79 ± 0.14	3.67 ± 0.21
Spleen (g/100 g B.W.)	0.35 ± 0.03	0.35 ± 0.05	0.35 ± 0.04	0.34 ± 0.04
Pancreas (g/100 g B.W.)	0.67 ± 0.10	0.84 ± 0.14	0.60 ± 0.03	0.72 ± 0.06
Kidney (g/100 g B.W.)	1.27 ± 0.06	1.29 ± 0.07	1.24 ± 0.05	1.31 ± 0.05
Lung (g/100 g B.W.)	0.87 ± 0.06	1.11 ± 0.11	0.89 ± 0.06	1.07 ± 0.10
Heart (g/100 g B.W.)	0.52 ± 0.02	0.52 ± 0.02	0.55 ± 0.03	0.57 ± 0.03
Brain (g/100 g B.W.)	1.53 ± 0.03	1.52 ± 0.04	1.56 ± 0.03	1.61 ± 0.05
Mesenteric fat (g/100 g B.W.)	0.61 ± 0.12	0.71 ± 0.16	0.47 ± 0.08	0.53 ± 0.07
Perirenal fat (g/100 g B.W.)	0.45 ± 0.10 ^ab^	0.88 ± 0.19 ^a^	0.60 ± 0.17 ^ab^	0.31 ± 0.08 ^b^
Epidydimal fat (g/100 g B.W.)	1.49 ± 0.23	2.30 ± 0.57	1.84 ± 0.50	1.14 ± 0.23
Plasma TG (mg/100 g mL)	141.3 ± 14.3 ^ab^	108.5 ± 7.5 ^ab^	144.8 ± 11.7 ^a^	102.0 ± 8.7 ^b^
Plasma TC (mg/100 g mL)	75.4 ± 5.1	66.2 ± 7.8	85.1 ± 2.4	62.2 ± 7.3
Plasma PL (mg/100 g mL)	36.8 ± 2.9	31.6 ± 3.3	39.0 ± 0.7	30.3 ± 3.0
Plasma glucose (mg/100 g mL)	86.8 ± 4.8	73.1 ± 11.2	88.8 ± 4.9	86.1 ± 5.0
Liver TG (mg/g)	53.8 ± 4.0	53.4 ± 3.8	42.4 ± 7.4	52.8 ± 6.1
Liver TC (mg/g)	3.9 ± 0.2	3.6 ± 0.1	3.8 ± 0.1	3.5 ± 0.2
Liver PL (mg/g)	4.0 ± 0.1	3.6 ± 0.1	3.8 ± 0.1	3.5 ± 0.2

B.W.: body weight, TG: triglyceride, TC: total cholesterol, PL: phospholipid. Data are shown as means ± SE and ab different letters represent significant difference at *p* < 0.05.

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
