# Peer review of "Evaluation of γ-oryzanol Accumulation and Lipid Metabolism in the Body of Mice Following Long-Term Administration of γ-oryzanol"

_nutrients, 2019, doi:10.3390/nu11010104_

Reviewer 1 Report

This study explored the relationship between the lipid-lowering effects of Oryzanol (OZ) and its accumulation using an obesity mice model. ORY is a mixed of bioactive compounds abundant in rice bran oil that gained a lot of attention for its lipid lowering activity. Its biological effects it has been thought to be due to its metabolite, ferulic acid. This study demonstrated instead that OZ itself can be found in blood stream and different organs at different amounts.  

The authors already published in 2016 the pharmacokinetic behavior of a single administration of rice brain oil. This study improved OZ pharmacokinetic study, well characterizing OZ itself distribution in different organs by HPLC/MS analysis after repeated administrations, and demonstrated OZ correlation with the plasma TG  decrease.  

Minor comments:

1) the total number of pregnant dams and pups. How many pups each dams gave birth?

2) the modality of mice feeding

3) the period of the last meal before the sacrifice. This is very important because it could change a lot the interpretation of the results.  

In addition:The introduction should be better focused on the topic of the work especially the first paragraph.

The references should be check carefully, because sometimes they did not fit with the concept in the text. 

Author Response

Response to Reviewer 1 Comments

Point 1: This study explored the relationship between the lipid-lowering effects of Oryzanol (OZ) and its accumulation using an obesity mice model. ORY is a mixed of bioactive compounds abundant in rice bran oil that gained a lot of attention for its lipid lowering activity. Its biological effects it has been thought to be due to its metabolite, ferulic acid. This study demonstrated instead that OZ itself can be found in blood stream and different organs at different amounts.

    The authors already published in 2016 the pharmacokinetic behavior of a single administration of rice brain oil. This study improved OZ pharmacokinetic study, well characterizing OZ itself distribution in different organs by HPLC/MS analysis after repeated administrations, and demonstrated OZ correlation with the plasma TG decrease.

Response 1: We wish to express our appreciation to the Reviewer 1 for the positive comments.

Point 2: the total number of pregnant dams and pups. How many pups each dams gave birth?

Response 2: We thank the reviewer for attentions to clarify the details. We revised the sentence “After birth, the dams were randomly assigned seven pups, and all offspring were then weaned at 3 weeks of age.” to “Each dam gave birth to 5-9 pups. After birth, seven pups were randomly assigned to each dam, and weaned at 3 weeks of age.” (Revised manuscript, P4L93).

Point 3: the modality of mice feeding

Response 3: Thank you for your comments. In this study, all mice were freely fed diets. We changed the sentence “Dams were housed individually in polycarbonate cages with free access to distilled water in a room at constant temperature (23 ± 1 oC) and humidity under a 12 h light/dark cycle.” to “Dams were housed individually in polycarbonate cages with free access to diet and distilled water in a room at constant temperature (23 ± 1 oC) and humidity under a 12 h light/dark cycle.” (Revised manuscript, P3L91). “At 3 weeks of age, male offspring were randomly separated into four dietary groups (n=9) and fed with test diets (AIN93G, SO-HF, RBO-HF, and HOZ-HF diets) for 8 weeks.” was changed to “At 3 weeks of age, male offspring were randomly separated into four dietary groups (n=9) and freely fed test diets (AIN93G, SO-HF, RBO-HF, and HOZ-HF diets) for 8 weeks.” (Revised manuscript, P4L94).

Point 4: the period of the last meal before the sacrifice. This is very important because it could change a lot the interpretation of the results.

Response 4: We do agree with the reviewer in this regard. The mice were fasted overnight (14 h) and then sacrificed. We revised the sentence “Mice were sacrificed by decapitation at 11 weeks of age, and their liver, brain, kidney, spleen, muscle, mesenteric fat, perirenal fat, epidydimal fat, and blood were collected.” to “Mice were fasted overnight (14 h) and then sacrificed by decapitation at 11 weeks of age, and their liver, brain, kidney, spleen, muscle, mesenteric fat, perirenal fat, epidydimal fat, and blood were collected.” (Revised manuscript, P4L96).

Point 5: The introduction should be better focused on the topic of the work especially the first paragraph. The references should be check carefully, because sometimes they did not fit with the concept in the text.

Response 5: Thank you for this valuable comment. As suggested by the Reviewer, we condensed the first paragraph of the introduction to focus specifically on the current work. The paragraph “Functional substances contained in plants such as vegetables and fruits are presumed to possess various bioregulatory functions. Compared to medicinal drugs, these substances have lower risks of side effects and secondary failures, so the use of these functional substances for the treatment and prevention of various diseases has attracted scientific attention. For example, it has been reported that catechins contained in tea reduce blood cholesterol level by suppressing the emulsification and absorption of lipids in the small intestine [1]. Also, it is well known that tocopherols and carotenoids demonstrate antioxidative properties, and therefore inhibit the oxidation of biological molecules (e.g., lipids) in vivo to prevent a variety of pathological conditions such as atherogenesis, diabetes, and aging [2,3]. To fully receive such functions, it is necessary to understand how these functional substances are absorbed and metabolized in vivo. However, studies regarding the absorption and metabolism of such substances remain limited, with the exception of a few reports on some well-known molecules (e.g., tocopherols, polyphenols, and certain carotenoids [4-6].” was changed to “Functional substances contained in plants such as cereals, vegetables and fruits are presumed to possess various bioregulatory functions. To fully receive such functions, it is necessary to understand how these functional substances are absorbed and metabolized in vivo. However, studies regarding the absorption and metabolism of such substances remain limited, with the exception of a few reports on some well-known molecules (e.g., tocopherols, polyphenols, and certain carotenoids [1-3].” (Revised manuscript, P1L33).

    We also checked the references and the concept in the text, and revised as follows; “Furthermore, in a different study using UV-HPLC, a relatively high UV absorption at 315 nm, corresponding to OZ, was detected from the brain 5 hours after a single administration of OZ [25].” was changed to “Furthermore, in a different study using UV-HPLC, a relatively high UV absorption at 315 nm, corresponding to OZ, was detected from the brain 1 hour after a single administration of OZ [22]. ” (Revised manuscript, P8L239). Also, reference numbers were revised throughout the manuscript.

   In addition to the above changes, “This research was funded in part by JSPS KAKENHI [grant number 17K19217]; and Public Interest Incorporated Foundation, Wakayama Industry Promotion Foundation (WIPF), Japan.” was changed to “This research was funded in part by JSPS KAKENHI [grant number 17K19217]; JSPS Core-to-Core Program (Advanced Research Networks) entitled “Establishment of international agricultural immunology research-core for a quantum improvement in food safety”; and Public Interest Incorporated Foundation, Wakayama Industry Promotion Foundation (WIPF), Japan.” (Revised manuscript, P9L286).

Reviewer 2 Report

This study analyzed OZ in plasma and organs to verify direct absortion of OZ and evaluated relationship between the presence of OZ in plasma and liver and physiological effect of OZ.

In other study, it is reported that orally administered OZ is hydrolyzed to ferulic acid. To see this, authors should have analyzed both ferulic acid and OZ.

Author Response

Response to Reviewer 2 Comments

Point 1: This study analyzed OZ in plasma and organs to verify direct absortion of OZ and evaluated relationship between the presence of OZ in plasma and liver and physiological effect of OZ.

Response 1: We wish to express our appreciation to the Reviewer 2 for the positive comments.

Point 2: In other study, it is reported that orally administered OZ is hydrolyzed to ferulic acid. To see this, authors should have analyzed both ferulic acid and OZ.

Response 2: Thank you for your comments. We understand the importance of understanding the metabolism of OZ to ferulic acid. Therefore, we are now developing an LC-MS/MS method for analysis of ferulic acid in plasma and organs. We are also carrying out a separate study to clarify the metabolism of OZ and wish to publish it in the future.

    In addition to the above changes, “This research was funded in part by JSPS KAKENHI [grant number 17K19217]; and Public Interest Incorporated Foundation, Wakayama Industry Promotion Foundation (WIPF), Japan.” was changed to “This research was funded in part by JSPS KAKENHI [grant number 17K19217]; JSPS Core-to-Core Program (Advanced Research Networks) entitled “Establishment of international agricultural immunology research-core for a quantum improvement in food safety”; and Public Interest Incorporated Foundation, Wakayama Industry Promotion Foundation (WIPF), Japan.” (Revised manuscript, P9L286).

Reviewer 3 Report

I have just a couple of suggestions. There is a strikingEffect on perirenal fat that needs to be discussed.  The extra fat in the diet is not well described. What constitutes the lipid in the high fat diet?

Author Response

Response to Reviewer 3 Comments

Point 1: I have just a couple of suggestions.

Response 1: We wish to express our appreciation to the Reviewer 3 for the positive comments.

Point 2: There is a strikingEffect on perirenal fat that needs to be discussed.

Response 2: Thank you for your advice. We are also interested in the significant change of the perirenal fat that was observed. The accumulation of OZ was confirmed in the perirenal fat as well as other organs, and thus we thought that OZ may have influenced the accumulation of fat. On the other hand, although accumulation of OZ could be confirmed in the mesenteric fat which is the same visceral adipose tissue, there was no significant difference in it. In order to elucidate the mechanism of these phenomena, we thought it is necessary to measure the expression levels of lipids and carbohydrate metabolism-related genes and proteins. However, since perirenal and mesenteric fat from mice were very limited, we unfortunately could not measure the gene expression in the perirenal fat. We are carrying out a separate study to clarify this matter and wish to publish it in the future. Upon consideration of the above, we added the sentence “Although accumulation of OZ was observed in mesenteric white adipose tissue which is also a visceral adipose tissue, no significant difference in mesenteric white adipose tissue weight was seen. Such difference is of interest, and further studies may be necessary to elucidate its mechanism.” (Revised manuscript, P5L178).

Point 3: The extra fat in the diet is not well described. What constitutes the lipid in the high fat diet?

Response 3: The content and type of lipid in each diet were as follows: AIN93G (containing 7% (w/w) fat as soybean oil), SO-HF (containing 20% (w/w) soybean oil), RBO-HF (containing 20% (w/w) normal rice bran oil), and HOZ-HF (containing 20% (w/w) rice bran oil with a high concentration of OZ). Such details of the diet composition are shown in Table 1.

   In addition to the above changes, “This research was funded in part by JSPS KAKENHI [grant number 17K19217]; and Public Interest Incorporated Foundation, Wakayama Industry Promotion Foundation (WIPF), Japan.” was changed to “This research was funded in part by JSPS KAKENHI [grant number 17K19217]; JSPS Core-to-Core Program (Advanced Research Networks) entitled “Establishment of international agricultural immunology research-core for a quantum improvement in food safety”; and Public Interest Incorporated Foundation, Wakayama Industry Promotion Foundation (WIPF), Japan.” (Revised manuscript, P9L286).

Round  2

Reviewer 2 Report

The reason for my decision of major revision is that ferulic acid should
be measured.
As stated in their manuscript, in other study, γ-oryzanol is converted
to ferulic acid and this ferulic acid is the active compound after
ingestion of γ-oryzanol.
while they hypothesized γ-oryzanol is not converted to ferulic acid and,
therefore, γ-oryzanol is the active compound.
However, they measured only γ-oryzanol.
To veryfy whether or not γ-oryzanol is the main active compound, they
should have measured both ferulic acid and γ-oryzanol.

Author Response

Response to Reviewer 2 Comments

Point 1: Point 1: The reason for my decision of major revision is that ferulic acid should be measured. As stated in their manuscript, in other study, γ-oryzanol is converted to ferulic acid and this ferulic acid is the active compound after ingestion of γ-oryzanol. while they hypothesized γ-oryzanol is not converted to ferulic acid and, therefore, γ-oryzanol is the active compound.

However, they measured only γ-oryzanol.

Response 1: Thank you for this valuable comment. First of all, our hypothesis never denies the activity of OZ metabolites including ferulic acid. We believe that the physiological functions from intake of OZ is exerted not only by OZ but also by its metabolites. Thus, in order to elucidate the conversion of OZ to ferulic acid, we agree that it is necessary to measure the concentration of ferulic acid. However, since plasma and organs samples excluding liver of mice were very limited, we unfortunately could not measure the ferulic acid level in their plasma and organs. Therefore, in order to evaluate the conversion of OZ to ferulic acid, we carried out additional experiments. Since the revision period was only 10 days, a single dose study of OZ was conducted. The study design is below.

----------------------------------------------------------------------------------------------------------------

    Male Sprague–Dawley rats were purchased from CLEA Japan. Rats were housed individually under conditions described above. After 18 hours fast, rats (n=4) received OZ (containing cycloartenyl ferulate, 24-methylenecycloartanyl ferulate, campesteryl ferulate, and β-sitosteryl ferulate) (181 mg/kg) dissolved in soybean oil (55.9 g/kg) via gastric intubation. After 6 hours of administration, blood was collected, and plasma was prepared from blood via centrifugation. Plasma (80 μL) was subjected to extraction and analysis of ferulic acid. The remaining plasma (20 μL from each rat) was combined and subjected to extraction and analysis of OZ described above. The protocol of this study was approved by the Center for Laboratory Animal Research, Tohoku University (Permit Number: 2018AgA-040).

    For extraction of ferulic acid, plasma (80 μL) was diluted in 400 μL of methanol containing 1% formic acid, vortexed, and subjected to centrifugation at 13,000 × g for 10 min at 4°C. The upper layer was collected, and the residue was similarly reextracted with 300 μL of methanol containing 1% formic acid. The upper layers were combined and evaporated under nitrogen gas. The dried extract was redissolved in 1.5 mL of methanol–water (1:9, v/v) containing 0.1% formic acid. This mixture was loaded onto an Oasis HLB cartridge (Waters, Massachusetts, U.S.A.) equilibrated with 0.1% formic acid aqueous solution. The cartridge was rinsed with 1.5 mL of 0.1% formic acid aqueous solution and ferulic acid was eluted with 1.5 mL methanol–water (9:1, v/v) containing 0.1% formic acid. The eluent was evaporated, dissolved in 1 mL acetonitrile–water (1:9, v/v), and subjected to HPLC-MS/MS analysis described below.

    Samples were analyzed with a 4000 QTRAP mass spectrometer (SCIEX) with a C8 column (InertSustain C8, 2 μm, 2.1 ID × 100 mm; GL Science, Tokyo, Japan) maintained at 40°C with a flow rate of 0.4 mL/min. Gradient elution was performed using a two-solvent system consisting of solvent A (water containing 0.1% formic acid) and solvent B (acetonitrile containing 0.1% formic acid). The gradient program was as follows: 0–3.5 min, 10–20% B linear; 3.5-5.5 min, 20–40% B linear; 5.5–6.0 min, 40% B. Ferulic acid was detected with MRM ion pairs described in Supplementary Table 1.

----------------------------------------------------------------------------------------------------------------

    By analyzing OZ and ferulic acid in plasma after a single administration of OZ to rats, it was found that the plasma concentrations of OZ and ferulic acid were similar; the total OZ level in plasma was 43.4 nM (cycloartenyl ferulate 16.4 ng/mL (27.1 nM), 24-methylenecycloartanyl ferulate 7.3 ng/mL (11.7 nM), campesteryl ferulate 1.9 ng/mL (3.3 nM), and β-sitosteryl ferulate 0.8 ng/mL (1.3 nM)), whereas that of ferulic acid was 36.1 ± 12.0 nM. Therefore, these results suggest the possibility that intact OZ, along with its metabolites (e.g., ferulic acid), is a biologically active form that exhibits physiological actions.

    Upon consideration of the above, we added the below to our manuscript;

 “2.6. Single oral administration of OZ to rats

    Male Sprague–Dawley rats were purchased from CLEA Japan. Rats were housed individually under conditions described above. After 18 hours fast, rats (n=4) received OZ (containing cycloartenyl ferulate, 24-methylenecycloartanyl ferulate, campesteryl ferulate, and β-sitosteryl ferulate) (181 mg/kg) dissolved in soybean oil (55.9 g/kg) via gastric intubation. After 6 hours of administration, blood was collected, and plasma was prepared from blood via centrifugation. Plasma (80 μL) was subjected to extraction and analysis of ferulic acid. The remaining plasma (20 μL from each rat) was combined and subjected to extraction and analysis of OZ described above. The protocol of this study was approved by the Center for Laboratory Animal Research, Tohoku University (Permit Number: 2018AgA-040).

    For extraction of ferulic acid, plasma (80 μL) was diluted in 400 μL of methanol containing 1% formic acid, vortexed, and subjected to centrifugation at 13,000 × g for 10 min at 4°C. The upper layer was collected, and the residue was similarly reextracted with 300 μL of methanol containing 1% formic acid. The upper layers were combined and evaporated under nitrogen gas. The dried extract was redissolved in 1.5 mL of methanol–water (1:9, v/v) containing 0.1% formic acid. This mixture was loaded onto an Oasis HLB cartridge (Waters, Massachusetts, U.S.A.) equilibrated with 0.1% formic acid aqueous solution. The cartridge was rinsed with 1.5 mL of 0.1% formic acid aqueous solution and ferulic acid was eluted with 1.5 mL methanol–water (9:1, v/v) containing 0.1% formic acid. The eluent was evaporated, dissolved in 1 mL acetonitrile–water (1:9, v/v), and subjected to HPLC-MS/MS analysis described below.

    Samples were analyzed with a 4000 QTRAP mass spectrometer (SCIEX) with a C8 column (InertSustain C8, 2 μm, 2.1 ID × 100 mm; GL Science, Tokyo, Japan) maintained at 40°C with a flow rate of 0.4 mL/min. Gradient elution was performed using a two-solvent system consisting of solvent A (water containing 0.1% formic acid) and solvent B (acetonitrile containing 0.1% formic acid). The gradient program was as follows: 0–3.5 min, 10–20% B linear; 3.5-5.5 min, 20–40% B linear; 5.5–6.0 min, 40% B. Ferulic acid was detected with MRM ion pairs described in Supplementary Table 1.” (Revised manuscript, P5L148).

    We also revised the sentences below;

“These results suggest the possibility that intact OZ is a biologically active form that exhibits physiological activities.” to “These results in combination with our additional data from a single oral administration test suggest the possibility that intact OZ, along with its metabolites (e.g., ferulic acid), is a biologically active form that exhibits physiological actions.” (Revised manuscript, P1L28),

 “2.6. Statistics” to “2.7. Statistics” (Revised manuscript, P5L175),

“This may suggest that intact OZ, along with its metabolites, is a biologically active form in vivo. In fact, in vitro tests have also suggested that OZ, in the intact form, demonstrates hypocholesterolemic activities [25]. However, further studies including the analysis of metabolites are necessary to prove this hypothesis. Also, since intact OZ was accumulated in organs other than the liver (Figure 4), further examination of these organs (e.g., fat) may be required to understand the relationship between the accumulation of OZ and its physiological effects in vivo.” to “This may suggest that intact OZ is a biologically active form in vivo. In fact, in vitro tests have also suggested that OZ, in the intact form, demonstrates hypocholesterolemic activities [25]. However, this consideration never denies the activity of OZ metabolites (e.g., ferulic acid). As a result of analyzing OZ and ferulic acid in plasma after a single administration of OZ to rats, it was found that the plasma concentrations of OZ and ferulic acid were similar; the total OZ level in plasma was 43.4 nM (cycloartenyl ferulate 16.4 ng/mL (27.1 nM), 24-methylenecycloartanyl ferulate 7.3 ng/mL (11.7 nM), campesteryl ferulate 1.9 ng/mL (3.3 nM), and β-sitosteryl ferulate 0.8 ng/mL (1.3 nM)), whereas that of ferulic acid was 36.1 ± 12.0 nM. Consequently, these results suggest the possibility that intact OZ as well as its metabolites (e.g., ferulic acid) are biologically active forms that exhibit physiological actions. Further studies including the analysis of metabolites such as ferulic acid and sterols are necessary to prove our hypothesis. Since intact OZ was also accumulated in plasma, liver, and other organs (Figure 4), further examination of these organs (e.g., fat and kidney) may be also required to understand the relationship between the accumulation of intact OZ, along with its metabolites, and their physiological effects in vivo.” (Revised manuscript, P9L290).

    Mr. Takumi Kokumai greatly contributed to analysis of ferulic acid. Therefore, he was added as an author of this study and to the “Author Contributions” section.
